# A Mutation Network Method for Transmission Analysis of Human Influenza H3N2

**DOI:** 10.3390/v12101125

**Published:** 2020-10-03

**Authors:** Chi Zhang, Yinghan Wang, Cai Chen, Haoyu Long, Junbo Bai, Jinfeng Zeng, Zicheng Cao, Bing Zhang, Wei Shen, Feng Tang, Shiwen Liang, Caijun Sun, Yuelong Shu, Xiangjun Du

**Affiliations:** 1School of Public Health (Shenzhen), Sun Yat-sen University, Guangzhou 510006, China; zhangch295@mail2.sysu.edu.cn (C.Z.); wangyh256@mail2.sysu.edu.cn (Y.W.); chenc57@mail2.sysu.edu.cn (C.C.); longhy7@mail2.sysu.edu.cn (H.L.); baijb@mail2.sysu.edu.cn (J.B.); zengjf7@mail2.sysu.edu.cn (J.Z.); caozch@mail2.sysu.edu.cn (Z.C.); zhangbing4502431@outlook.com (B.Z.); shenw3@mail.sysu.edu.cn (W.S.); tangf35@mail2.sysu.edu.cn (F.T.); liangshw5@mail2.sysu.edu.cn (S.L.); suncaijun@mail.sysu.edu.cn (C.S.); shuylong@mail.sysu.edu.cn (Y.S.); 2Key Laboratory of Tropical Disease Control, Ministry of Education, Sun Yat-sen University, Guangzhou 510006, China

**Keywords:** influenza virus, mutation network, transmission, phylogenetic analysis

## Abstract

Characterizing the spatial transmission pattern is critical for better surveillance and control of human influenza. Here, we propose a mutation network framework that utilizes network theory to study the transmission of human influenza H3N2. On the basis of the mutation network, the transmission analysis captured the circulation pattern from a global simulation of human influenza H3N2. Furthermore, this method was applied to explore, in detail, the transmission patterns within Europe, the United States, and China, revealing the regional spread of human influenza H3N2. The mutation network framework proposed here could facilitate the understanding, surveillance, and control of other infectious diseases.

## 1. Introduction

Yearly epidemics of seasonal influenza cause an enormous disease burden around the globe, and they are responsible for approximately three to five million annual cases of severe illness and 290,000 to 650,000 respiratory deaths [1]. The annual spread of seasonal influenza virus is driven by the continuous evolution of the virus itself as a result of the immune system [2,3]. For the effective surveillance and control of influenza, the World Health Organization (WHO) has established an international network of collaborating laboratories to monitor the spread and antigenic variation of influenza viruses [4]. An understanding of the spatial spread of human influenza is of critical importance for targeted surveillance and precise vaccine recommendations, ultimately leading to better control of human influenza.

The global circulation pattern of human influenza virus, especially for the H3N2 subtype, was well characterized by many previous studies, where the East and Southeast Asia regions are central to the emergence of new variants, while the United States may play an important role in their global spread [3,5,6,7]. However, regional transmission patterns for migration hubs such as Europe, the United States, and China are less studied, despite their importance for both regional and global surveillance and the control of seasonal influenza.

Several methods were proposed to study the spatial transmission of seasonal influenza using phylogenetic theory [8,9,10]. With the rapid development of network theory and its wide application in interdisciplinary fields [11,12,13], researchers are also exploring its potential in the transmission analysis of infectious diseases [5,14,15,16]. Compared with traditional phylogeographic methods, network presentation relies on fewer assumptions and aligns with the intrinsic nature of pathogen networks [14,17,18]. This field is still in its infancy, and the full potential of network theory in the study of infectious diseases needs to be explored.

In this study, we developed a mutation network framework to study the spatial transmission of human influenza H3N2 using sequence data. We further applied this framework to an in-depth transmission analysis in Europe, the United States, and China.

## 2. Materials and Methods

### 2.1. Simulations

The regional expansion and transmission of seasonal influenza H3N2 were simulated according to a realistic scenario. The expansion of human influenza H3N2 within a region was simulated using a specific transmission model with a periodic transmission rate involving a cycle of one year to simulate the seasonality of human influenza virus.
(1)ΔI(t+1)=λ(t)ΔI(t)
$$\lambda \left(t\right)=\mu [1+\sin\left(\frac{1}{26}\pi t\right)]$$
where ΔI is the number of new infected cases at time t, and λ(t) is the transmission rate, formulated using a periodic function, with a scaling factor μ, which was fixed at 0.75. Infected people were expected to recover within 1 week. As a result, ΔI is also the number of infected cases at time t. The transmission events between regions (six regions: A, B, C, D, E, and F) was calculated on the basis of a predefined transmission intensity.
(2)$$N{\left(t\right)}_{j\to i}=\{\begin{array}{c} \;0,\;\Delta I{\left(t\right)}_{j}TI_{j\to i}<1\\ \Delta I{\left(t\right)}_{j}TI_{j\to i},\;\Delta I{\left(t\right)}_{j}TI_{j\to i}\geq 1 \end{array}$$
where $N{\left(t\right)}_{j\to i}$ is the number of transmission events from region j to region i, $\Delta I{\left(t\right)}_{j}$ is the number of infected cases at time t for region j, and $TI_{j\to i}$ is the predefined transmission intensity from region j to region i (see Appendix A). The relative interregional transmission intensity was then calculated as the interregional transmission intensity divided by the total transmission intensity across all regions. The transmission events for a specific region were calculated as the sum of transmission events from all other regions. Each year, one specific region was selected as the source region with one initial infection as a function of the trunk proportion (0.35, 0.25, 0.15, 0.1, 0.08, and 0.07 for regions A, B, C, D, E, and F, respectively). The simulation was run weekly and lasted 20 years. The genetic sequences were simulated as a function of a predefined mutation probability [19] and a mutation rate of 6.7 × 10^−3^ per site and year [20]. The mutation probabilities between A and C, A and G, A and T, C and G, C and T, and G and T were set to 0.1, 0.5, 0.07, 0.04, 0.25, and 0.04, respectively [19]. The whole transmission process was recorded, including the sequences. For each region, 200 sequences were sampled every year, while 100 repeated simulations were carried out.

### 2.2. Bedford Data

A dataset from Bedford et al. [7] was used in this study for validation. In total, 4004 hemagglutinin (HA) sequences of the human influenza H3N2 virus, which were evenly sampled from nine regions in the period from January 2000 to April 2012, were included in the dataset. The number of passengers on interregional flights from their study was also used. For comparison, the interregional migration rate and trunk proportion for each region, obtained from phylogeographic analysis, were also used.

### 2.3. Nucleotide Sequence Data

Nucleotide HA sequences of human influenza H3N2 from Europe and the United States (US) between April 2008 and November 2019, and from China between September 2010 and April 2018 were downloaded from the Global Avian Influenza Data Sharing Initiative (GISAID) website. Overall, 22,867, 19,113, and 1802 sequences were used for Europe, the US, and China, respectively. Sequences were aligned with MUSCLE v3.7 using default settings [21]. Undetermined nucleotides or gaps were replaced with the highest-frequency nucleotides at the corresponding position on the basis of the top 10 closest sequences measured by Hamming distance. A phylogenetic tree was constructed using FastTree 2 with default settings [22]. Outliers on the phylogenetic tree and redundant sequences from the same place and season were removed from the dataset. The summer season was defined as the period from April to September, whereas the winter season was defined as the period from October to March of the next year. In total, 100 sampling datasets were constructed by randomly selecting five and 20 sequences in the summer and winter seasons, respectively, for each selected country in Europe and for each division from the US Department of Health and Human Services (HHS). Similarly, two and 10 sequences were randomly selected in the summer and winter seasons from each province or region in China. In situations involving a lack of sequences, they could all be used.

### 2.4. Mutation Network Construction

#### 2.4.1. Initial Network

The initial network was constructed with nodes representing strains and links representing pairs of strains with a mutation probability no more than a predetermined threshold, which was defined as the 10th percentile of a collection of the highest mutation probabilities from each strain with their closest relative (Figure 1A, initial network). Baseline mutation probabilities among A, T, G, and C were extracted from pairs of sequences with single-nucleotide differences in the corresponding data. Mutation probabilities between pairs of strains with different numbers of mutations were calculated as the product of probabilities of single mutations.

#### 2.4.2. Fully Connected Network

The initial network could not be fully connected if strains were too far apart. As a result, in the second step, an isolated subnetwork was connected to its closest subnetwork via the link with the highest probability. This process was iteratively repeated until the network was fully connected. Consequently, an undirected and weighted network was obtained, with the weight representing the mutation probability between two nodes (Figure 1A, connected network).

#### 2.4.3. Directed Network

In order to define the direction of links, the earliest strain was selected as the root. The direction was then defined on the basis of the shortest paths from the root to all other strains. The shortest path was extracted using the Dijkstra algorithm, and the weights were transformed using the original probability with a negative logarithmic function. Then, a partial directed network was constructed, and directions for the remaining links were defined according to the following principles: (1) the direction went from the strain with a higher mutation probability from the root to the strain with a lower mutation probability from the root; (2) the direction between two strains with the same mutation probability from the root was defined as going from the strain with the earlier collection date to the strain with the later collection date; (3) a circular structure was not allowed, with the direction being flipped if one appeared. Finally, a directed and weighted network was constructed (Figure 1A, directed network).

### 2.5. Transmission Analysis

In the mutation network, each node has a geographic attribute. In order to quantify the transmission pattern, random walks were undertaken in the mutation network. For a better representation of the starting point, a random node was selected on the basis of its Pagerank value [23]. Then, the next move was determined as a function of the weight, representing the mutation probability to its child nodes. This process was repeated until a terminal node (a node without children) was attained. A transmission event was recorded upon a change in location (Figure 1A, transmission analysis). The whole process was repeated 10,000 times to obtain a relative transmission intensity between two regions (i.e., the total number of transmission events between two regions normalized by the number of all transmission events).

### 2.6. Trunk Analysis

The betweenness centrality (BC) was used to quantify the “trunkness” of nodes in the network. Due to the single-trunk structure of the mutation network for human influenza H3N2, the BC for nodes located in the middle of the trunk was systematically amplified. To correct this bias, a new measure, named trunkness, was constructed by counting the number of the shortest paths passing through a node between a randomly chosen strain and each terminal strain in the network. This was repeated 1000 times, and the relative proportion of trunkness for all nodes was used as an indicator of proportion in the trunk (Figure 1A, trunk analysis).

### 2.7. Statistical Analysis

Pearson correlation analysis was used in this study. All analysis results were carried out using a two-sided test with 0.05 as the significance level.

We used Python (version 3.7, Python Software Foundation, https://www.python.org/) and its extension libraries Pandas and Numpy for data cleaning and processing, Networkx for network analysis, Matplotlib and Basemap for mapping, and Gephi0.9.2 (https://gephi.org) for network visualization [24].

## 3. Result

### 3.1. Mutation Network Framework and Simulation Validation

The mutation network was constructed using large-scale influenza sequences through a series of processes and, eventually, a directed, weighted, and connected network was generated (Figure 1A). A range of algorithms, including random walk, shortest path, Pagerank, and BC, were used to study interregional transmission patterns and the source probability (probability of being the origin) of the human influenza H3N2 virus (see Section 2 for more details).

In order to test the effectiveness of the mutation network framework in inferring interregional transmission patterns of influenza and the source probability from genetic sequences, a well-calibrated simulation was carried out (see Section 2 for more details). The results show that the analysis using the mutation network could correctly retrieve the interregional transmission patterns and source probabilities. The correlations between estimated and predefined values were as high as 0.85 and 0.97 for relative transmission intensity and trunk proportion, respectively (Figure 1B,C).

### 3.2. Transmission Analysis for Bedford Data

We used the dataset published by Bedford et al. [7] to demonstrate the mutation network framework. Following the procedures outlined in Section 2, a fully connected, weighted, and directed network was constructed with 3946 nodes and 95,364 edges (see Figure 2A). As can be seen, the evolution process of human influenza H3N2 followed a single trunk structure with relatively short side branches. The out degree of nodes (number of child nodes) followed a power law distribution [25], suggesting the presence of a few hub strains with several children but many strains with a few children (see Appendix A).

Nine geographical regions were defined for the dataset from Bedford et al. [7]: US/Canada, South America, Europe, India, North China, South China, Japan/Korea, Southeast Asia, and Oceania (Figure 2B). The results show that there was close communication between US/Canada and Europe, while regions from Asia were tightly connected (Figure 2B). We determined a high correlation between our proposed transmission matrix and that from Bedford et al. (Pearson correlation 0.81, *p* < 0.001; Figure 2C). According to current knowledge, flight traffic is the major factor driving global influenza transmission [6,26]. The correlation between travel flow and transmission intensity in our mutation network was also high (Pearson correlation 0.76, *p* < 0.001; Figure 2D).

### 3.3. Trunk Analysis for Bedford Data

The trunk in the mutation network represents the main spreading route during global migration of human influenza H3N2. A greater proportion of a node in the trunk denotes its greater importance in the global spread of human influenza H3N2. The results show that viruses from South and North China occupied the largest proportion of the trunk (18.5% and 9.9%, respectively), followed by Europe (15.9%), and the United States/Canada (13.2%) (Figure 3B). This result was highly correlated with the result from the study by Bedford et al. [7] (0.69, *p* = 0.04) (Figure 3C).

### 3.4. Transmission Analysis for Europe, the United States, and China

Using the mutation network framework proposed herein, we determined more detailed transmission patterns for specific regions. In this study, we chose Europe, the United States, and China for further evaluation, as they play important roles in the global transmission of human influenza H3N2. To reduce the bias caused by sequence data from different regions, we carried out our analysis using mutation networks constructed with balance-sampled sequences from different countries or regions (see Section 2 for more details). We employed the 11 European countries (Sweden, Norway, the United Kingdom, the Netherlands, Germany, France, Spain, Austria, Italy, Russia, Switzerland) with sufficient data, and we used the 10 divisions of the US Department of Health and Human Services (HHS). Furthermore, in order to avoid sampling biases during the summer, when influenza activity is low in Europe and the United States, these strains were not counted. For China, we only focused on the southern provinces and northern regions with sufficient data in the summer season (Beijing, Tianjin, and Shandong), while we used the three regions proposed by Yu et al. for the winter season [27].

The results show that selected European countries contributed substantially to the trunk (Figure 4A). Regions from the northwest US occupied a larger proportion of the trunk, which indicates their important role in the spread of human influenza H3N2 (Figure 4B). In China, Guangdong, was the most important region in the spread of human influenza H3N2 during the summer (Figure 4C). During the winter, all three geographic regions played important roles in the spread with a comparable proportion in the trunk (Figure 4D).

## 4. Discussion

In this study, we proposed a sequence-based mutation network framework and applied it to study the transmission of human influenza H3N2. Our study indicates the feasibility of network theory in the study of infectious diseases, allowing for more detailed transmission information to be ascertained within the areas of Europe, the United States, and China, which will be helpful for the targeted surveillance and control of human influenza H3N2.

The mutation network framework was validated using a simulation, and results obtained using the mutation network were consistent with those obtained using a traditional phylogenetic method. Our results in the United States and China were also consistent with epidemiological data from the literature [28,29,30]. In Europe, previous studies showed that there is a phenomenon of spread from Western to Eastern Europe, whereas the trend between Northern and Southern Europe is not significant [31,32]. In this study, due to the availability of sequences, almost all countries in this study were from Western Europe (except Russia) and, indeed, no clear trend was observed within these countries with comparable trunk proportions. Regions from the northeast US play important roles in the spread of human influenza H3N2. Surveillance in these regions should be strengthened. South China plays an important role in the spread of human influenza H3N2 in the summer, whereas three regions contribute equally during the winter season, which indicates the rapid circulation of human influenza H3N2 during this period in China. North China does not typically experience seasonal influenza in the summer; however, our results indicate that Beijing and Tianjin also play important roles in the spread of human influenza H3N2 during this period. These two cities are within the capital economic region and are deemed major transport hubs in China. As municipalities, these two cities have strengthened their surveillance system, and, as a result, they may better capture new variants of the human influenza H3N2 virus.

Our study also had several limitations. Recently, a median-joining network was used to study the evolutionary relationship of the novel coronavirus (SARS-CoV-2) [33]. However, this method was criticized due to its distance-based phenetics and lack of direction [34]. There were also other explorations applying network theory in the study of infectious diseases, but most were based on simple assumptions [5,35,36]. The mutation network proposed here was a directed network with mutation probabilities representing the weights of links; however, the network did not include intermediate or hidden nodes, which may represent unsampled strains. Furthermore, the transmission and trunk analyses were based on either Pagerank or BC with random walks; thus, other measures should be carefully evaluated [37]. Moreover, although we tried to balance the temporal and spatial distribution of the sequence data through our sampling approach, its effect was limited. This drawback may be exacerbated when fewer sequence data are available. With the development of sequencing technologies, especially portable equipment with lower cost, more sequences will be accumulated in the future; therefore, mutation networks are expected to play an increasingly important role in the study of infectious diseases.

## Figures and Tables

**Figure 1 viruses-12-01125-f001:**
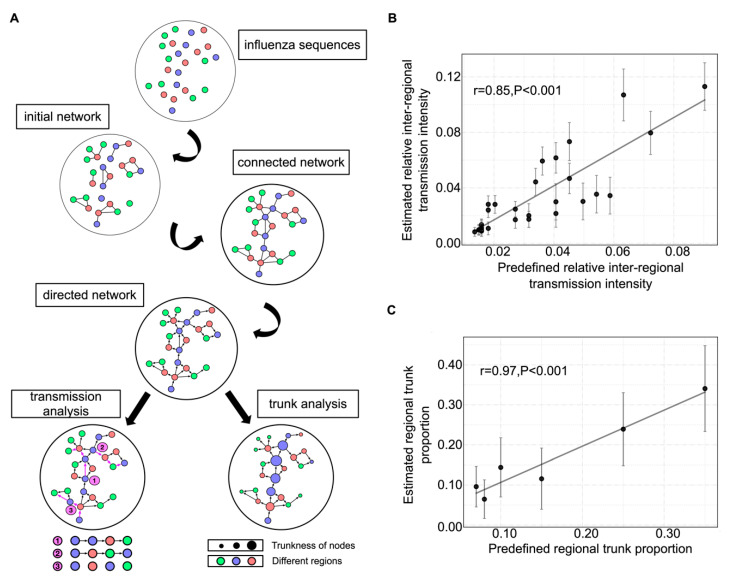
Mutation network framework and simulation validation. (**A**) Flowchart of the mutation network framework. Regions are differentiated using colors. Trunkness is the corrected betweenness centrality of nodes (see Section 2 for details). (**B**) Correlation between predefined and estimated relative interregional transmission intensity. (**C**) Correlation between predefined and estimated trunk proportion. The 95% confidence intervals are also given.

**Figure 2 viruses-12-01125-f002:**
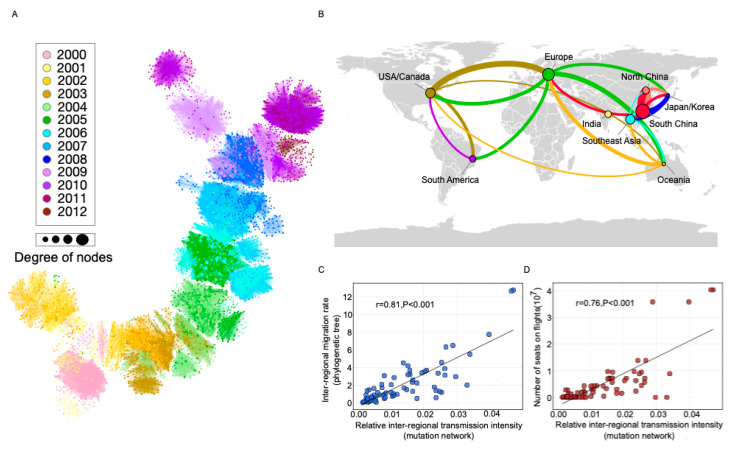
Global transmission and correlation analysis using the mutation network of human influenza H3N2. (**A**) Mutation network of human influenza H3N2. The color of the circle represents the time attribute of the node, while its size represents the degree of the node. (**B**) Global transmission network of human influenza H3N2 according to mutation network framework. The size of the circle represents the proportion of the region in the trunk of the mutation network. Transition lines between regions are colored on the basis of their origin. The thickness of the line represents the intensity of transmission and, for clarity, only lines with transmission intensity greater than 0.02 are shown. (**C**) Correlation between the relative transmission intensity from our mutation network and the results from the study by Bedford et al. [7]. (**D**) Correlation between the relative transmission intensity from our mutation network and the number of flight passengers between different regions.

**Figure 3 viruses-12-01125-f003:**
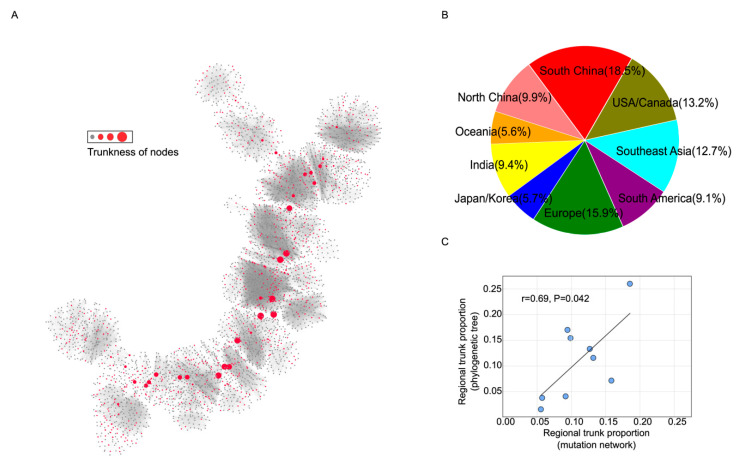
Trunk and correlation analysis using the mutation network of human influenza H3N2. (**A**) Trunkness of nodes in the mutation network. The trunkness of a node was defined by counting the number of shortest paths passing through a node between a randomly chosen strain and each terminal strain (node without children) in the network. Red circles represent the nodes contributing to the trunk, whereas their size represents the extent of the contribution. Gray circles represent the nodes not contributing to the trunk. (**B**) Proportion of each region in the trunk of the mutation network. (**C**) Correlation between the proportion in the trunk from the mutation network and that from the study by Bedford et al. [7].

**Figure 4 viruses-12-01125-f004:**
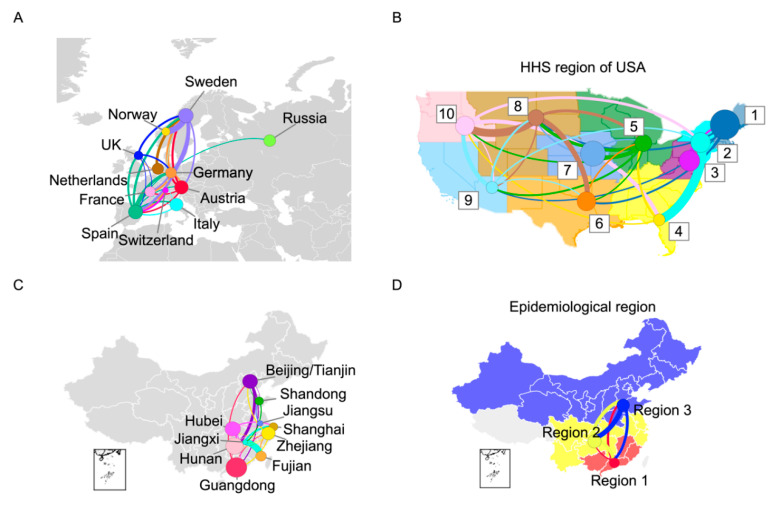
Transmission patterns in Europe, the United States, and China. (**A**) Regional transmission pattern in Europe. (**B**) Regional transmission pattern in the United States. Numbers 1-10 represent the 10 divisions of the US Department of Health and Human Services (HHS). (**C**) Transmission pattern in China for the summer season. (**D**) Transmission pattern in China for the winter season. The size of the circle represents the proportion in the trunk of the mutation network. Transition lines between regions are colored on the basis of their origin. The thickness of the line represents the transmission intensity (incomparable between subfigures) and, for clarity, only lines with interregional transmission intensity greater than 0.01 are shown.

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
