# Peer review of "A Mutation Network Method for Transmission Analysis of Human Influenza H3N2"

_viruses, 2020, doi:10.3390/v12101125_

Round 1

Reviewer 1 Report

I appreciate the care the authors took with revising the manuscript.

Reviewer 2 Report

I thank the authors for their extensive revisions and response to my comments. The authors have made a serious attempt to address my concerns, and I am generally happy with the analysis and results. There are, however, some sections that I found confusing or unjustified in ways that need to be addressed. Some of these might only require a revision of the description, or better justification. For others I am not sure.

The English could use some editing. “Could rely on less assumptions” should be “could rely on fewer assumptions”. “Need to be explored” should be “needs to be explored”.

Regarding the simulation described in section 2.1, I do not see a growth rate. You are not modelling the growth rate with a periodic function, but rather modelling the number of infected people as a periodic function. The growth rate should be reflected by the derivative of I rather than the value of I. I do not understand lambda as the growth index as lambda varies with the sin of t and the derivative of I varies with the cos of t. (dI/dt = I (1-mu) cos(omega t)/omega where omega equals pi/26). I also do not understand the cure and death rate because these also vary sinusoidally which does not seem appropriate. If I(t) is the number of infected people at time t, then is I(t)_j the number of people infected in region j at time t? Why are the number of transmitted cases inversely proportional to the population of the source region? It would seem larger regions are more likely to be sources.

The construction of the initial network is confusing. Firstly, I assume you are linking strains with the mutational probability no less than a predetermined threshold. I do not understand the term “collection of lowest mutation probabilities for each strain”. If you are looking at each strain with its closest relative, what then makes it a collection? That would seem to give you a single number, which makes the 10th percentile confusing. I am also unhappy with the idea of mutation probabilities as calculated here. The model seems to represent the situation where each site has an independent probability of differing. Of course these probabilities are not independent. If the two strains are more divergent, all of the sites are more likely to be mutated, and so the probability of mutations at different sites are correlated. Why not just consider evolutionary divergence, which can also take into account base specific mutation probabilities, although including something like gamma distributed rates might be more important. I can imagine that the author's approach might be acceptable if better justified, but I am not sure that is the case.

I would call it a terminal node rather than the dangling node.

It would have been useful to better justify the trunkness analysis.

Just a suggestion, but figure 1B seems to have some interesting structure. It might be interesting to see if there is some characteristic shared by those links where the transmission intensity is underestimated relative to overestimated.

I think the correlation in figure 3C is not highly correlated. The correlation coefficient of .69 means that it explains less than half the variance. It seems that if you were to remove the China point there would be essentially no correlation or maybe an inverse correlation. I would go back to say moderately correlated. New paragraph

In conclusion I like this work and hope the authors are able to make the revisions necessary for it to get the reception it deserves.

Author Response

This manuscript is a resubmission of an earlier submission. The following is a list of the peer review reports and author responses from that submission.

Round 1

Reviewer 1 Report

In the revised manuscript, authors were showed mutation network framework of influenza viruses could facilitate the understanding and ultimately the surveillance and control of other infectious diseases. It is possible to apply and useful for understanding of global circulation pattern of mutated influenza viruses from human as well as other species including avian and swine. Minor comment that authors need to describe that sequence-based mutation network framework consistent with serological or epidemiological evidence of circulation influenza viruses.

Reviewer 2 Report

The authors describe an interesting approach to an important problem. It is obvious that disease transmission follow directed networks, and so it makes sense to formulate the problem in this manner.

My main problem is that the authors provide no evidence that the method is successful, or reason to believe that the method described is effective. There are many different steps and it is not clear the efficacy of these steps, where they succeed, where they break down. This is especially true as networks are not between the sequences that are observed but rather the (unknown) sequences that are transmitted, with lots of internal nodes that are missing. The true network goes through these internal nodes, with the observed sequences emerging as branches from these nodes. I am not saying that the method is obviously wrong, rather that it is unproven and therefore I do not know what to do with the results of the analysis.

I am also concerned about the results shown in figure 3C, which seems to indicate that there is a discrepancy between the results of the mutation network and the phylogenetic tree. The correlation between these two measures, whether statistically significant, seems inadequate to develop confidence in the method.

I would be happier if there were data on known transmission pathways where the model could be tested. Alternatively, I would want some demonstration, potentially involving synthetic data, that indicates that the model provides useful results.

Reviewer 3 Report

In this manuscript by Chi Zhang et al., the authors used a mutation network framework to describe ten years of evolution of the HA sequence of the human H3N2 viruses. According to the authors, this method allowed them to correlate this evolution with international travel data.

Main findings

The authors used the HA-nucleotide sequences of 4004 H3N2 viruses, from the [april2008-nov2019] period for Europe and US, and from the period [sept2010-april2018] for China. These sequences were downloaded from the GISAID site.

The sequences were then progressively connected to each other to build a tree, and to identify transmission between distinct regions of the world. 

While the subject is interesting, it is unclear to what extent this study reveals new information. The added value of the network theory is not at all clear, and it is even difficult to understand in what the method used by the authors is different from the classical phylogenetics methods. It is not even convincing that the method used any tool or principle that is specific to network theory.

Major remarks

Lines 47-50. The authors used the dataset from Bedford (2015). However they then said that sequences from the [2008-2019] or [2010-2018] periods were downloaded from the GISAID site. This is not coherent, since the Bedford article was published in 2015. Further, the authors used “4004 HA sequences of human H3N2”, but a search on GISAID from april 1st 2008 to November 30th 2019 identifies only 3029 isolates with complete HA sequence. 

Lines 52-53. The authors should precise that they worked on nucleotide sequence data.

Lines 69-72. This is not clear at all. And what is the meaning of the values A-G:0.5, etc?

Lines 82-88. This is not clear. 

Figure 1a. The nodes and edges are mapped onto a plane, but what is the meaning of the coordinates (x, y) of each point? Are they (x and y) related to time, to genetic distance, or to other parameters, or more simply is their distribution optimized for readability? It would be helpful to zoom-in on a very small region of this map, in order to view the relationships between a small number of isolates (with the name, location an sampling dates of the isolates).

Further, the authors should try to better interpret the map in Figure 1a: for instance it seems that most of the 2001-2002 isolates are not connected to the 2003 and 2004 isolates. Rather it would seem that the 2003-2004 isolates evolved from only a very small subset of sequences from 2002. If it is correct, this may be interesting and deserves some discussion.While Figure 1b is easy to understand, the meaning of Fig 1C is unclear and probably not enough interpreted. 

Lines 131-139 and Figure 2. There is absolutely no introduction to this section, and its rationale is not explained. Further, there is no data to support the interpretation of the authors. With no clear data, no one can judge the validity of the claims made by the authors (this remark also applies to Figure 2). In figure 2, there is no direct connection between Asia and America.

Lines 149-155 and Figure 3. It is not at all clear how the authors compared their results with those of Bedford. More specifically, what exactly do the points represent in Fig 3C? (same question also applies to Fig. 2B). 

Lines 174-180 and Fig. 4. The data and their interpretation are not convincing. The role of Sweden is surprising, and the authors propose no hypothesis to try to explain that (could it result from some bias in the sampled sequences, or from a parameter of the analysis?). Further, there is no real information or useful interpretation in sentences like “regions from the northwestern, northeastern and south play important roles in the spread of human influenza H3N2 in the in the United States”. 

Minor remarks

Lines 31-32 seasonal influenzaLine 42… mutation network framework to study…Line 57… Hamming distance (from Richard Hamming)

Line 61…. By randomly selecting

Line 77…. Weighted network was obtained..

Line 131…nine geographical regions were defined…

English grammar should be reviewed in depth. Several sentences are grammatically incorrect (for instance lines 157-159…)